# Morphometric MRI Evaluation of Three Autografts Used in Anterior Cruciate Ligament Reconstruction in Athletes

**DOI:** 10.3390/jfmk8010014

**Published:** 2023-01-23

**Authors:** Christos K. Yiannakopoulos, Georgios Theotokatos, Iakovos Vlastos, Nikolaos Platon Sachinis, Elina Gianzina, Georgios Kalinterakis, Olympia Papakonstantinou

**Affiliations:** 1School of Physical Education and Sport Science, National and Kapodistrian University, 17237 Athens, Greece; 22nd Department of Radiology, Attikon University Hospital, National and Kapodistrian University, 12462 Athens, Greece

**Keywords:** anterior cruciate ligament, autograft, magnetic resonance imaging, morphometry

## Abstract

The purpose of the present study was to quantify the morphometric characteristics of three tendon autografts (hamstring tendons (HT), quadriceps tendon (QT), and patellar tendon (PT)) used in anterior cruciate ligament (ACL) reconstruction. For this purpose, knee magnetic resonance imaging (MRI) was obtained in 100 consecutive patients (50 males and 50 females) with an acute, isolated ACL tear without any other knee pathology were used. The level of the physical activity of the participants was determined using the Tegner scale. Measurements of the tendons’ dimensions (PT and QT tendon length, perimeter, cross-sectional area (CSA), and maximum mediolateral and anteroposterior dimensions) were performed perpendicular to their long axes. Higher values were recorded as regards the mean perimeter and CSA of the QT in comparison with the PT and the HT (perimeter QT: 96.52 ± 30.43 mm vs. PT: 63.87 ± 8.45 mm, HT: 28.01 ± 3.73 mm, F = 404.629, *p* < 0.001; CSA QT: 231.88 ± 92.82 mm^2^ vs. PT: 108.35 ± 28.98 mm^2^, HT: 26.42 ± 7.15 mm^2^, F = 342.415, *p* < 0.001). The length of the PT was shorter in comparison with the QT (53.1 ± 7.8 vs. 71.7 ± 8.6 mm, respectively, *t* = −11.243, *p* < 0.001). The three tendons showed significant differences in relation to sex, tendon type, and position as regards the perimeter, CSA, and the mediolateral dimensions but not for the maximum anteroposterior dimension.

## 1. Introduction

The anterior cruciate ligament (ACL) is one of the main knee joint ligamentous stabilizers, and loss of its function following a traumatic event often results in instability, especially in movements involving rotation or a sudden change of direction [1,2,3]. Chronic ACL rupture often precipitates damage to the other stabilizing and anatomical elements of the knee joint, such as the menisci and the articular cartilage, resulting in secondary knee osteoarthritis [4,5,6].

An ACL rupture is the most common severe knee ligament injury, responsible for 200,000 cases per year in the United States, both in the general population and in athletes [7]. The main goal of ACL reconstruction surgery (ACLR) is the replacement of the torn ligament, usually with an autograft, with the primary aim of restoring the stability and kinematics of the knee joint, facilitating a return to previous physical activity [8].

The outcome of an ACLR depends on many factors related to the type of the graft, including its dimensions, cross-sectional area (CSA), biomechanical strength, correct placement, and graft fixation strength [9]. Hamstring tendon autografts with a diameter less than or equal to 7.5 mm are twice as likely to fail when compared with grafts with a diameter greater than or equal to 8 mm [9].

Preoperative knee imaging can be used to investigate whether the dimensions of the selected graft meet the graft size requirements. Previous morphological studies have evaluated the relative differences between the tendon grafts but usually only in one location [10].

However, to the best of our knowledge, there are no data on the variation in the dimensions of the tendons in relation to their length. Multiple measurements theoretically provide a more detailed preoperative assessment of the potential tendon grafts. Our study focused on comparing the dimensions of the three most used tendon autografts in the knee (patellar tendon, hamstrings tendons, and quadriceps tendon) in multiple positions along their course. The working hypothesis is that the autograft tendon dimensions around the knee joint show significant variations not only between them but also along their length. The main purpose of the study was to describe a methodology that will assist in the preoperative autograft tendon selection process in patients scheduled for ACL reconstruction surgery.

## 2. Materials and Methods

### 2.1. Design and Participants

The morphometric analysis of four knee tendons was performed in 50 men and 50 women (overall 100 subjects) using 3D magnetic resonance imaging (MRI) of the knee. The study population included skeletally mature athletic men (n = 50) and women (n = 50), aged 20 to 40 years, without any significant injury, symptomatology, knee pathology, or operations in the lower limbs. The level of physical activity of the participants was determined with the Tegner scale [11,12]. The Tegner scale provides a standard method of evaluating work and sports activity and includes 11 categories (Table 1), selecting the one that best describes the patient’s activity level. The study involved people with a score on the Tegner scale ranging from 4 to 9. The study was approved by the Research Ethics-Biology Committee of the School of Physical Education and Sport Science of the National and Kapodistrian University of Athens (approval number: 1104/13-03-2019).

### 2.2. Magnetic Resonance Imaging Technique

All participants underwent knee MRI using a 3 Tesla MRI (Magnetom Skyra, Siemens Healthcare GmbH, Erlangen, Germany) with a dedicated eight-channel phased-array knee coil. All exams were performed in the supine position with the knee in extension according to the standard protocol. The field of view was widened to include the quadriceps tendon and the patellar tendon along their entire length. A 3D Proton Density SPACE sequence with spectral fat suppression was acquired in the sagittal plane with a TR/TE 1200/28 ms matrix and field of view of 16 × 16 cm, with 0.5 mm thick sections. Reconstructions of the images in the axial and coronal planes were obtained. This sequence was isotopic, and the small slice thickness allowed for 3D reconstruction. The examination protocol also included coronal T1 and sagittal images as part of the routine examination. The DICOM files were transferred, stored, and analyzed using a picture archiving and communication system (PACS). Axial reconstructions perpendicular to the longitudinal axis of the measured tendon were performed.

The patellar tendon (PT) was evaluated at four levels: (1) at the lower pole of the patella, (2) at the tibial tubercle level prior to its attachment, and (3) and (4) at two equidistant locations between the proximal and distal attachments. The quadriceps tendon (QT) was assessed at the upper pole of the patella and proximally at three or four more locations with a 1 cm distance between them. The hamstring tendons (HT) were evaluated in a section perpendicular to their course at the level of the knee joint. In each position, the perimeter, the CSA, the anteroposterior diameter (height), and the transverse diameter (width) were measured. The anteroposterior dimensions of the patellar and the quadriceps tendons were measured at the widest point of the tendon. Finally, the lengths of the patellar and the quadriceps tendons were also measured (Figure 1a, b). The length of the hamstring tendon was not included in the study because our initial MRI measurements showed low reproducibility due to the fact of their oblique course and, second, because the proximal extent of the magnetic field was limited to 10 cm proximal to the upper patellar pole.

### 2.3. Statistical Evaluation

For data processing and analysis, SPSS for Windows, version 28 (IBM Corp., Armonk, NY, USA) was used. The measurements were initially evaluated for homogeneity and symmetry (skewness and kurtosis) and were suitable for parametric processing and analysis. One-way ANOVA for repeated measures was used to compare the dependent variables for each point of measurement of the same tendon (patellar and quadriceps) at different positions, as well as for comparison between the different points on the two tendons (patellar and quadriceps). Multiple comparisons were made between the four tendon measurement positions using Bonferroni correction. Finally, comparisons were made between men and women in each tendon per position using two-way ANOVA. All tests were two-way, and the statistical significance level was set at *p* < 0.05.

The reliability of the measurements was evaluated using the intraclass correlation coefficient (ICC). The ICC is a reliability indicator used in the analysis of the repeatability of measurements both between evaluators (interrater) and between repeated measurements by the same researcher (intrarater). In the present study, the ICC was calculated, and the 95% confidence interval based on the average (k = 3), absolute agreement (absolute agreement), and 2-way mixed effects model. Values less than 0.5 indicate low reliability, from 0.5 to 0.75 moderate, from 0.75 to 0.90 good, and greater than 0.90 exceptional reliability [13]. The measurements were performed by a musculoskeletal radiologist, with 20 years of experience, who served as the reference, an orthopedic surgeon, and a physiotherapist.

## 3. Results

The results of the inter- and intrarater reliability evaluation were conducted using the MRIs of 30 individuals, which are presented in the Appendix A (Table A1 and Table A2). The length and CSA of the PT and the QT and the CSA of the semitendinosus and gracilis tendons (HT) were evaluated. The 95% confidence intervals are presented in the respective tables, as is the statistical significance of the measurements.

The study was performed in a sample of healthy, athletic men and women without any previous injury or surgical operation in the lower limbs. The mean age and Tegner scale scores of the participants, overall and according to sex, are presented in Table 2. Overall, the mean age of the participants was 28.64 ± 5.32 years, and the mean Tegner scale score was 6.11 ± 1.46. There was no significant difference between the two genders for these two variables.

The quadriceps tendon was longer than the patellar tendon (71.70 ± 8.63 and 53.14 ± 7.76 mm, respectively, with t = −11.243, *p* < 0.001) (Table 3). There was no significant difference between the two genders for the patellar (t = 0.056, *p* = 0.955) or the quadriceps tendon (t = −0.704, *p* = 0.487).

The one-way ANOVA showed a significant difference between the PT, QT, and HT for the tendon perimeter (F = 40.629, *p* < 0.001). The CSA showed significant differences between the three tendons (F = 34.415, *p* < 0.001) (Table 4, Figure 2). Two-way ANOVA was used to examine the interaction between the tendon and gender for the perimeter, with no statistically significant result (F = 3.093, *p* = 0.086; Table 4, Figure 3). With respect to the tendon and gender for the CSA, statistically significant results were found (F = 8.373, *p* = 0.006).

A two-way ANOVA was also used to examine the interaction between the four measurement sites (1, 2, 3, and 4) of the patellar and the quadriceps tendons (Table 5, Figure 4, Figure 5, Figure 6 and Figure 7). As regards the CSA, there was a significant interaction (F = 51.420, *p* < 0.001) and significant effect for the tendon (F = 292.046, *p* < 0.001) and the position independently (F = 79.734, *p* < 0.001).

## 4. Discussion

In the present study, it was shown that the morphometric characteristics (perimeter, cross-sectional area, width, and height) of the three tendons around the knee joint commonly used as autografts in ACLR varied significantly depending on the location of the measurement. The tendons showed significant differences according to sex, type of tendon, and position examined as regards their perimeter, cross-sectional area, and width.

The perimeter and the cross-sectional area of the HT were compared with those of the QT and the PT and found to be significantly smaller, as expected. In the ACL reconstruction surgery, only part of the QT or the PT, usually one-third of it, was harvested and used for the ACL substitution. The length of the PT was smaller than that of the QT (53.14 ± 7.76 vs. 71.70 ± 8.63 mm, *p* < 0.001). This result is to be expected, since the patellar tendon is limited between the lower pole of the patella and the tibial tubercle, whereas the quadriceps tendon blended proximally into the quadriceps muscle. The average cross-sectional area of the PT was approximately 46.57% of the cross-sectional area of the QT (108.35 ± 28.98 versus 231.88 ± 92.82 mm^2^, respectively). In another study [10], the cross-sectional area of the PT was 46.49% of the cross-sectional area of the QT (33.2 ± 7.3 vs. 71.4 ± 10.5 mm^2^, respectively). The cross-sectional area of the HT differed from the present study because they were measured intraoperatively. In our study, the HT tendons were measured at the level of the joint line (26.42 ± 7.15 mm^2^ in our study versus 55.3 ± 8 mm^2^). The cross-sectional area and the perimeter of the quadriceps tendon was significantly higher compared to the other two autografts (patellar and hamstring tendons). The cross-sectional area of the HT was only 11% of the quadriceps tendon and 46% of the patellar tendon (26.42 ± 7.15 versus 231.88 ± 59.64 and 108.35 ± 25.32 mm^2^, respectively, with a *p*-value < 0.001). The CSA of the three common autografts varied significantly, and there was not a strong correlation between them [10].

Another finding of the present study was the variation in the dimensions of the tendons along their length. At the patellar insertion site, the dimensions of the quadriceps and the patellar tendons were higher and decreased proximally and distally. This observation is important when MRI measurements are performed preoperatively in only one measurement location, as is the case in most studies, leading to the over- or underestimation of the tendon autograft dimensions.

There were no significant differences between the men and women in the dimensions of the PT and the QT at different locations, probably because there was no significant difference in the mean height between the two groups. Despite this, higher values were recorded in the males regarding the perimeter, cross-sectional area, maximum transverse, and maximum anteroposterior diameter. However, there was no statistically significant difference in the total perimeter and cross-sectional area of the HT. This may be due to the morphology of the specific tendons, i.e., their cylindrical shape [14].

Most studies did not evaluate simultaneously all three tendons in the same individuals, and the majority performed measurements at a single location, not allowing to draw conclusions regarding the variation in the tendons’ dimensions along their entire length [10,15,16,17]. The exception is a study by Camarda et al. [16], which, however, did not measure the cross-sectional area of the tendons.

The 3D SPACE MRI sequence that we used, with a slice thickness of 0.5 mm, allowed us to perform measurements with high accuracy due to the clear delineation of the tendon boundaries. In other studies, a 1.5 Tesla or 3 Tesla MRI was used but without the same protocol [15,17,18,19,20,21,22,23]. We evaluated the reliability of our measurements comparing the examiners with different experience and scientific backgrounds, showing excellent reliability, both between examiners and between the same examiner’s measurements. The reliability and accuracy of MRI measurements have been reported in several previous studies. One study found that MRI measurements differed by only 0.5 mm in 75% of patients when preoperative MRI measurements were compared with a QT autograft [18], which has been confirmed by other studies as well [20,22]. The reliability and validity of MRI as a tool for the preoperative prediction of the size of autografts has been studied using different sequences [21,24]. Hanna et al. [24] concluded that MRI is a valid method of predicting preoperatively the dimensions of the hamstring tendons with a total ICC of 0.977, which is in line with the results of the present study.

The preoperative prediction of the autograft size using MRI is important in the preoperative planning of an operation, guiding the selection of the most suitable autograft for the ACL reconstruction procedure. When the autograft size is insufficient, especially when hamstring tendons are used, the procedure becomes more complex due to the need to augment [25] or replace the hamstring autograft with another auto- or allograft. The additional surgical trauma increases the morbidity, length, and cost of the operation. The addition of allograft strands to a small-diameter hamstring autograft does not lead to a successful outcome, probably due to the poor healing potential [26]. In a systematic review, it was shown that the correlation between the preoperative MRI measurements and intraoperative graft sizing was highly or very highly positive only for the patellar and the quadriceps tendon, respectively, while it was negligible–highly positive for the semitendinosus-only tendon and negligible–moderately positive for the gracilis-only tendon [27].

The present study has various strengths and weaknesses. The study was performed in a rather homogenous population of young athletes, males and females; a 3 Tesla MRI and a 3D SPACE sequence were used; and all evaluations were performed at a level perpendicular to the respective tendons to avoid measurement bias, which may occur when the MRI images are oblique rather than perpendicular to the tendon. On the other hand, the results of the study cannot be extrapolated in the pediatric population, due to the variation in the tendons’ dimensions during development, or the elderly population due to the effect of aging and reduced physical activity on the tendons’ dimensions. The dimensions of the hamstring tendons were evaluated only at one location (i.e., at the level of the knee joint), although proximally the tendons tended to be thinner. This may overestimate the final hamstring tendons’ CSA. Furthermore, no intraoperative tendon measurements were performed, and the clinical outcome of the ACLR was not taken into consideration. As a result, no correlation analysis between the morphometric properties and the clinical results is available.

In summary, we found that there were statistically significant morphometric differences between the dimensions of the three tendon autografts used in the ACL reconstruction, and there was a clear variation in the morphometric properties along their length. The morphometric characteristics of the quadriceps tendon were superior compared to the other tendons. Finally, we found statistically significant differences between the two sexes in the dimensions of the tendons at different locations for the QT and PT; however, there was no statistically significant difference between the two sexes in the dimensions of the HT.

## Figures and Tables

**Figure 1 jfmk-08-00014-f001:**
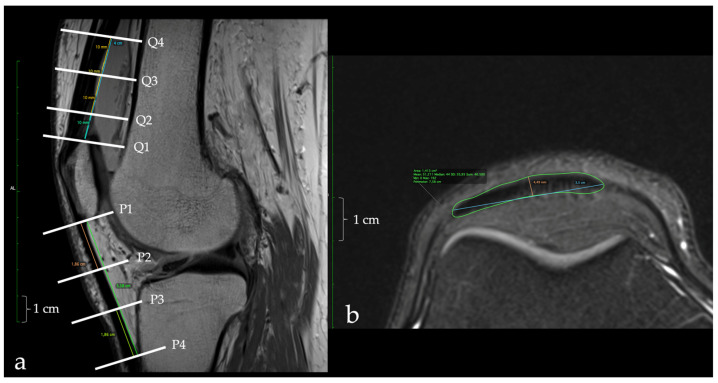
(**a**,**b**) MRI evaluation of the dimensions of the patellar and the quadriceps tendons. All measurements were performed at a level perpendicular to the respective tendon (**a**). The measurements in the quadriceps tendon were performed at the patellar attachment of the tendon and at 3 sites proximally with a 1-cm distance between them (Q1-Q4). In the patellar tendon, 4 measurements were performed at the proximal (P1) and the distal (P4) tendon attachment and in two positions between those measurement sites (P2 and P3). In (**b**), an example of the measurements in the axial plane is shown. The mediolateral and anteroposterior patellar tendon length, as well as the cross-sectional area of the patellar tendon are shown.

**Figure 2 jfmk-08-00014-f002:**
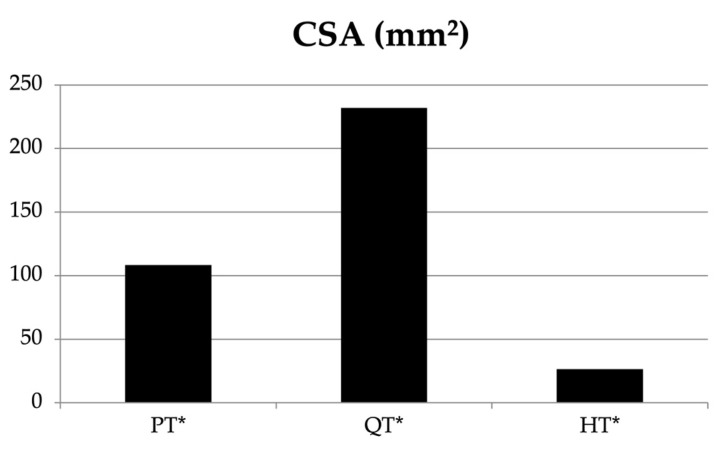
Representation of the cross-sectional area (CSA) of the patellar tendon (PT), the quadriceps tendon (QT), and the hamstring tendons (HT). * Statistically significant.

**Figure 3 jfmk-08-00014-f003:**
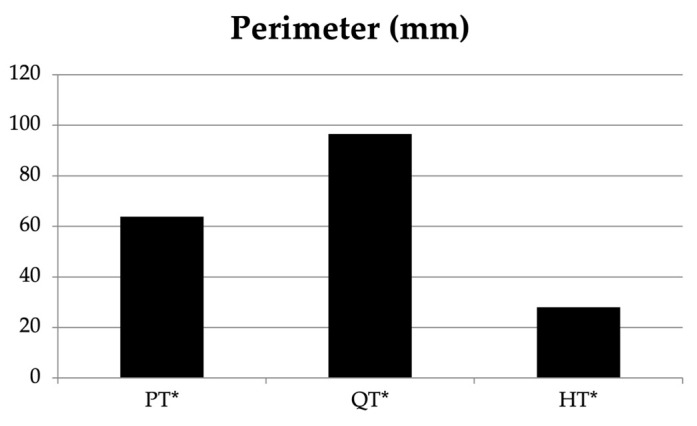
Bar graph showing the perimeter of the patellar tendon (PT), the quadriceps tendon (QT), and the hamstring tendons (HT). * Statistically significant.

**Figure 4 jfmk-08-00014-f004:**
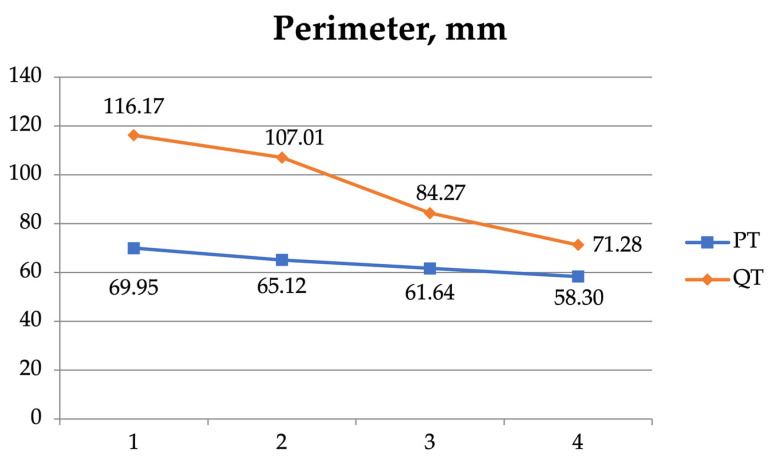
The perimeter of the patellar tendon (PT) and the quadriceps tendon (QT) at the 4 measurement sites.

**Figure 5 jfmk-08-00014-f005:**
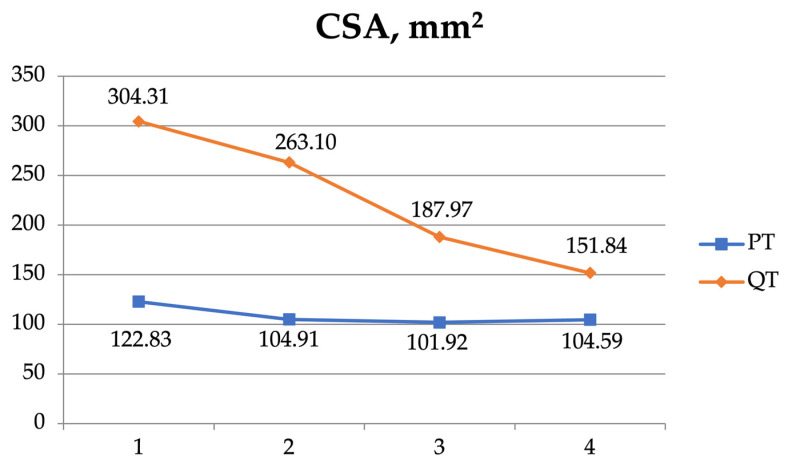
The cross-sectional area (CSA) of the patellar tendon (PT) and the quadriceps tendon (QT) at the 4 measurement sites.

**Figure 6 jfmk-08-00014-f006:**
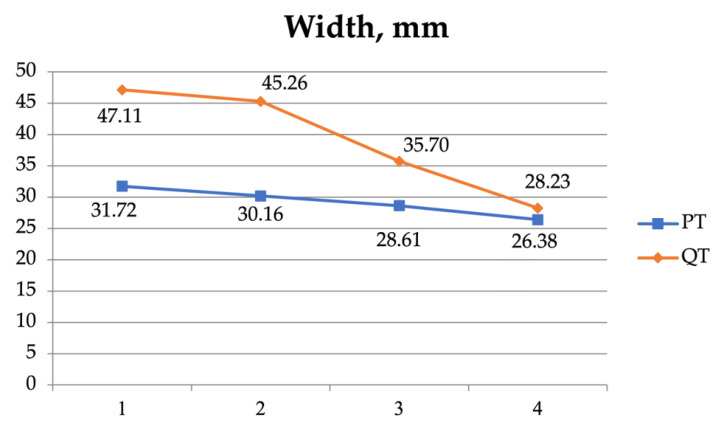
The width of the patellar tendon (PT) and the quadriceps tendon (QT) at the 4 measurement sites.

**Figure 7 jfmk-08-00014-f007:**
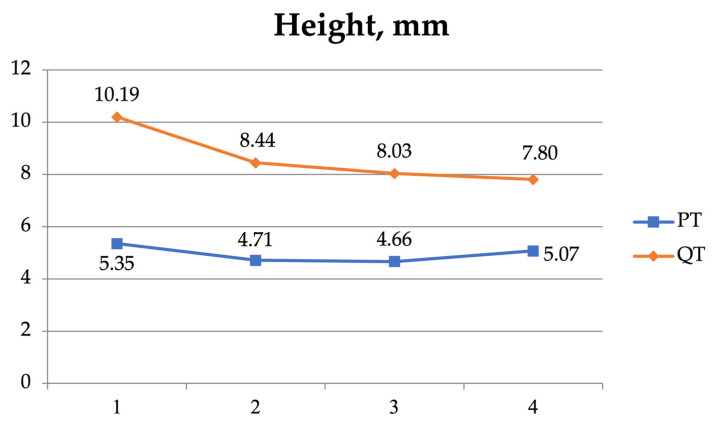
The anteroposterior diameter (height) of the patellar tendon (PT) and the quadriceps tendon (QT) at the 4 measurement sites.

**Table 1 jfmk-08-00014-t001:** The Tegner activity level scale.

Level	Activity
Level 10	Competitive sports—soccer, football, rugby (national or international level)
Level 9	Competitive sports—soccer, football, rugby (lower divisions), ice hockey, wrestling, gymnastics
Level 8	Competitive sports—racquetball, bandy, squash, badminton, track and field athletics (jumping, etc.), down-hill skiing
Level 7	Competitive sports—tennis, running, motocross, speedway, handball, basketball, Recreational sports- soccer, football, rugby, bandy, ice hockey, squash, racquetball, jumping
Level 6	Recreational sports—tennis and badminton, handball, racquetball, down-hill skiing, jogging at least 5 times per week
Level 5	Competitive sports—bicycling, cross-country skiing, Work- heavy labor (construction, building, forestry), Recreational sports- jogging on uneven ground at least twice weekly
Level 4	Recreational sports—bicycling, cross-country skiing, jogging on even ground more than twice weekly, Work- moderately heavy labor (e.g. truck driving, etc.)
Level 3	Competitive and recreational sports—swimming, walking in rough forest terrain, Work- light labor (nursing, etc.)
Level 2	Work—light labor, Walking on uneven ground possible, but impossible to back-pack or hike
Level 1	Work—sedentary (secretarial, etc.), Walking on even ground possible
Level 0	Sick leave or disability pension because of knee problems

**Table 2 jfmk-08-00014-t002:** Descriptive characteristics of the participants: age, height, and Tegner scale scores. SD, standard deviation; N = 100.

Variable	Males (Ν = 50)	Females (Ν = 50)	Overall	Statistical Significance
	Average ± SD	Average ± SD	Average ± SD	
Age	28.66 ± 5.10	28.62 ± 5.58	28.64 ± 5.32	*p* > 0.05
Height	175.87 ± 5.44	172 ± 4.98	173.64 ± 5.91	*p* > 0.05
Tegner Scale Score	6.20 ± 1.54	6.02 ± 1.39	6.11 ± 1.46	*p* > 0.05

**Table 3 jfmk-08-00014-t003:** Mean length of the patellar and the quadriceps tendon according to sex. SD, standard deviation.

Variable	Males (Ν = 50)	Females (Ν = 50)	Overall
	Average ± SD	Average ± SD	Average ± SD
Patellar Tendon Length, mm	53.22 ± 6.96	53.06 ± 8.82	53.14 ± 7.76
Quadriceps Tendon Length, mm	70.68 ± 7.62	72.85 ± 9.79	71.7 ± 8.63

**Table 4 jfmk-08-00014-t004:** Measurement of the Perimeter and the Cross-Sectional Area (CSA) for the Patellar Tendon (PT), the Quadriceps Tendon (QT), and the Hamstring Tendons (HT), overall, and according to sex.

Variable		Males (Ν = 50)	Females (Ν = 50)	Overall
		Average ± SD	Average ± SD	Average ± SD
Perimeter, mm	PT	66.38 ± 6.44	60.4 ± 4.94	63.87 ± 6.96
	QT	100.64 ± 17.5	87.07 ± 13.2	96.52 ± 17.4
	HT	28.35 ± 3.82	27.61 ± 3.73	28.1 ± 3.73
CSA, mm^2^	PT	116.76 ± 23.3	98.08 ± 19.7	108.35 ± 25.3
	QT	251.28 ± 62.1	195.53 ± 35.8	231.88 ± 59.6
	HT	26.97 ± 7.81	25.79 ± 6.53	26.42 ± 7.15

**Table 5 jfmk-08-00014-t005:** Comparison between the patellar and the quadriceps tendons’ dimensions at four different measurement sites distal and proximal to the distal and the proximal patellar pole, accordingly. CSA, cross-sectional area; SD, standard deviation.

Variable	Position of Evaluation	Patellar Tendon	Quadriceps Tendon	*p*-Value
		Average ± SD	Average ± SD	
Perimeter (mm)	1	69.95 ± 7.50	116.17 ± 23.51	<0.001
	2	65.12 ± 7.05	107.01 ± 28.29	<0.001
	3	61.64 ± 6.44	84.27 ± 20.71	<0.001
	4	58.30 ± 6.47	71.28 ± 18.46	<0.001
Cross-Sectional Area (mm^2^)	1	122.83 ± 36.22	304.31 ± 79.05	<0,001
	2	104.91 ± 22.74	263.10 ± 77.47	<0.001
	3	101.92 ± 21.65	187.97 ± 55.80	<0.001
	4	104.59 ± 23.31	151.84 ± 51.01	<0.001
Width (mm)	1	31.72 ± 3.09	47.11 ± 8.05	<0.001
	2	30.16 ± 3.34	45.26 ± 8.84	<0.001
	3	28.61 ± 3.13	35.70 ± 8.64	<0.001
	4	26.38 ± 2.99	28.23 ± 6.79	0.062
Height (mm)	1	5.35 ± 1.39	10.19 ± 13.73	0.026
	2	4.71 ± 0.96	8.44 ± 1.42	<0.001
	3	4.66 ± 0.83	8.03 ± 1.43	<0.001
	4	5.07 ± 0.82	7.80 ± 1.55	<0.001

## Data Availability

Data are available from the authors upon request.

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
