# Peer review of "Morphometric MRI Evaluation of Three Autografts Used in Anterior Cruciate Ligament Reconstruction in Athletes"

_jfmk, 2023, doi:10.3390/jfmk8010014_

Round 1

Reviewer 1 Report (Previous Reviewer 2)

Dear authors,

Many thanks for your efforts for revising the manuscript.

Author Response

Thank you, Sir, for your valuable help.

Reviewer 2 Report (New Reviewer)

Introduction, 1st paragraph needs rephrase: "Chronic ACL rupture regardless of surgical reconstruction leads often precipitates damage to the other stabilizing and anatomical elements"

Introduction needs to be shorter. Extensive references to other studies could be better reserved for Discussion, like the following :  "In a study, 1480 patients with a hamstring tendon autograft were evaluated 2 to 15 49 years postoperatively and its was found that grafts with a diameter less than or equal to 50 7,5 mm, were twice as likely to fail when compared with grafts with a diameter greater 51 than or equal to 8 mm"

Author Response

Comment 1. Introduction, 1st paragraph needs rephrase: "Chronic ACL rupture regardless of surgical reconstruction leads often precipitates damage to the other stabilizing and anatomical elements"

Reply 1. Corrected. Thank you.

Comment 2. Introduction needs to be shorter. Extensive references to other studies could be better reserved for Discussion, like the following :  "In a study, 1480 patients with a hamstring tendon autograft were evaluated 2 to 15 49 years postoperatively and its was found that grafts with a diameter less than or equal to 50 7,5 mm, were twice as likely to fail when compared with grafts with a diameter greater 51 than or equal to 8 mm"

Reply 2. The length of the introduction was reduced, as well as the sentence you mentioned. Thank you for the valuable comments.

Reviewer 3 Report (New Reviewer)

congratulations to the authors, very interesting article

Author Response

Thank you, Sir, for your valuable help.

Reviewer 4 Report (New Reviewer)

The study explores a very important topic in the orthopedic field. The methods are clear and well presented, as are the results. However, I believe that the discussion can be better explored, especially if you want to "describe a methodology which will assist in the pre-operative autograft tendon selection process in patients scheduled for ACL reconstruction surgery" as reported in the introduction. 

I would add an image and a reference about the Tegner scale.

Please use only abbreviations, after they are introduced in the text.

Specify in the text why you didn't consider the length of HT

Author Response

  • The study explores a very important topic in the orthopedic field. The methods are clear and well presented, as are the results. However, I believe that the discussion can be better explored, especially if you want to "describe a methodology which will assist in the pre-operative autograft tendon selection process in patients scheduled for ACL reconstruction surgery" as reported in the introduction. 
    • The discussion was improved to better delineate our study’s goal, i.e., to assist in the preoperative autograft selection process.
  • I would add an image and a reference about the Tegner scale.
    • A table with the Tegner scale and 2 references were added.
  • Please use only abbreviations, after they are introduced in the text.
    •  
  • Specify in the text why you didn't consider the length of HT
    • Comment added, lines 165-167

Reviewer 5 Report (New Reviewer)

thanks for submitting this well-performed and interesting study.

minor comments from my side:

-please be sure that references are relevant and up to date

-please be sure that English grammar is perfect

-intro: 31-35 please delete nonrelevant parts, stay focused on the topic and explain why is this study needed

-statistical section: please report sample size calculation

-please start the discussion by reporting your main findings

-it would be interesting to know how much the MRI measurement correlates with the actual dimension of the graft. which grafts were used for  ACLR? were the grafts measured during surgery? please discuss this.

Author Response

  • -please be sure that references are relevant and up to date
    • Already checked. All recent references are included.
  • -please be sure that English grammar is perfect
    • Improved by a native English speaker
  • -intro: 31-35 please delete nonrelevant parts, stay focused on the topic and explain why is this study needed
    • Several lines were deleted and the introduction and discussion became more focused.
  • -statistical section: please report sample size calculation
    • We did not do a sample size calculation because we did not compare different individuals but the different tendon dimensions were compared within the same subject. We used 100 MRIs to have enough data to describe the variation of the patellar and quadriceps tendon dimensions along the length. Statistical significance, when the 3 tendon sources were compared, was achieved with only 20 cases. In all cases the quadriceps tendon is longer and thicker compared to the patellar tendon.
  • -please start the discussion by reporting your main findings
    •  
  • -it would be interesting to know how much the MRI measurement correlates with the actual dimension of the graft. which grafts were used for  ACLR? were the grafts measured during surgery? please discuss this.
    • Our study was morphometric only, using preoperative MRIs. We did not routinely compare the preoperative with the intraoperative measurements because this was not within the scope of the study. Several other studies have evaluated preoperative imaging with intraoperative measurements with excellent correlation. We have recently completed a study comparing the preoperative dimensions of the quadriceps tendon and the patellar tendon with intraoperatively measured dimensions in patients undergoing TKR with excellent correlation.

Round 2

Reviewer 4 Report (New Reviewer)

Thank you for your corrections

This manuscript is a resubmission of an earlier submission. The following is a list of the peer review reports and author responses from that submission.

Round 1

Reviewer 1 Report

Dear Authors, the manuscript is interesting even if in a niche it could give intriguing tips to those who approach a serious problem such as the reconstruction of the ACL. On the impact of the manuscript on the literature I leave it to the editor to decide, as regards the mere adaptability to the publication, I would like you to address these concerns:

Before the outcome in the abstract, put the characteristics of inclusion and exclusion of the population

Among other things, what gender proportion exists in the results? How many are the participants? What characteristics do they have?

29 Chronic ACL rupture? Provide a picture of risk factors if anything .. neuromuscular activation, perhaps anatomical or hormonal certainly linked to the phenomenon of "dynamic valgus knee"…

I suggest a sort of “The ACL injuries commonly occur in noncontact situations during direction changes, cutting maneuvers, or during landing after jump. Few prospective studies have investigated biomechanical risk factors of ACL injury, but it seems that the injury is linked to poor neuromuscular control of the knee stabilizing muscles and to the dynamic valgus condition to which the knee can be subjected even in contexts of contact sports .. "Ref: http://doi.org/10.7752/jpes.2020.05342

34-36 this is not necessarily the only indication .. I would suggest removing

In the methods you don't already put the results of the selection, you put inclusion and exclusion criteria .. have you removed statistically "non-normal" subjects… too “tall” for certain heights .. too “short” for certain heights? BMI? Athletic or sporting experience?

83 all procedure is operator dependent? You have made a triplecheck of the measurements between experts (also report it in the outcome measurement paragraph as well as in Stat An ..

In the results I would like to place the tables on reliability in the supplementary materials .. in the manuscript I would leave in a discursive way the good reliability of the approach .. I suggest this to ensure greater clarity

Figures lack standard deviation or at least SE .. they are critical to the type of illustrative manuscript for readers you have drafted. However, unfortunately the overall results are not very intriguing if it is not possible to evaluate the gender differences .. so I recommend to put male and female otherwise I would remove the figures.

Author Response

  • Before the outcome in the abstract, put the characteristics of inclusion and exclusion of the population.

Implemented the proposed change.

  • Among other things, what gender proportion exists in the results? How many are the participants? What characteristics do they have?

Lines 69-81: In section 2.1 the sample size and participants characteristics can be seen. The gender proportion is 1:1 (Overall 100 subjects, 50 men and 50 women). Table 1 shows the descriptive characteristics of the participants.

  • 29 Chronic ACL rupture? Provide a picture of risk factors if anything .. neuromuscular activation, perhaps anatomical or hormonal certainly linked to the phenomenon of "dynamic valgus knee"…

This is certainly a point, but we think that it is not directly related to the main purpose of the study, i.e. to describe the morphometry of the 3 tendons used for ACL reconstruction. A table like the one you are kindly proposing may be included in the discussion section.

  • I suggest a sort of “The ACL injuries commonly occur in noncontact situations during direction changes, cutting maneuvers, or during landing after jump. Few prospective studies have investigated biomechanical risk factors of ACL injury, but it seems that the injury is linked to poor neuromuscular control of the knee stabilizing muscles and to the dynamic valgus condition to which the knee can be subjected even in contexts of contact sports .. "Ref: http://doi.org/10.7752/jpes.2020.05342

Implemented the proposed change. Had to change the in-text citation and bibliography. Thank you.

  • 34-36 this is not necessarily the only indication .. I would suggest removing

Implemented the proposed change

  • In the methods you don't already put the results of the selection, you put inclusion and exclusion criteria .. have you removed statistically "non-normal" subjects… too “tall” for certain heights .. too “short” for certain heights? BMI? Athletic or sporting experience?

In results section we state that: “The measurements were initially evaluated for homogeneity and symmetry (skewness and kyrtosis) and were suitable for parametric processing and analysis”. We did not have to exclude cases

  • 83 all procedure is operator dependent? You have made a triplecheck of the measurements between experts (also report it in the outcome measurement paragraph as well as in Stat An ..

Yes. Only 1 experienced technician performed the MRI. The comparison of the measurements between the evaluators was performed by a blinded to the study statistician.

  • In the results I would like to place the tables on reliability in the supplementary materials .. in the manuscript I would leave in a discursive way the good reliability of the approach .. I suggest this to ensure greater clarity

Table 1 has been moved to the appendix. The order of the Tables’ names and in text has also been changed.

  • Figures lack standard deviation or at least SE .. they are critical to the type of illustrative manuscript for readers you have drafted. However, unfortunately the overall results are not very intriguing if it is not possible to evaluate the gender differences .. so I recommend to put male and female otherwise I would remove the figures.

Thank you for your valuable comment.  We chose to include the figures in order to be easier for the reader to visualize the variation of each dependent variable (CSA, perimeter etc.) between the different location of measurement. There were significant differences not only between the tendons, but within the same tendon as well.

Reviewer 2 Report

Authors performed morphometric evaluations using MRI for 3 autologous tendon grafts in ACLR.

This study is interesting, however there are major issues to be addressed.

The purpose of the study is not definite.

The hypothesis is not suggested clearly.

This study only assessed MRI measurements without clinical outcomes. So, there is no correlation analysis between the morphometric results and clinical results.

For the improved understanding, figures measuring the parameters in MRI should be added for readers. 

What is the clinical importance of this study? 

Line 96-105. Please suggest references for the reasons why authors assessed in this way. 

line 235. we performed evaluated..? 

What are the limitations of this study?

Author Response

  • The purpose of the study is not definite.

The main purpose of the study is to describe a methodology which will assist in the preoperative autograft tendon selection process in patients scheduled for ACL reconstruction surgery. (Lines 74-76)

  • The hypothesis is not suggested clearly.

The working hypothesis is that the autograft tendon dimensions around the knee joint show significant variations not only between them but also along their length.( Lines 72-74)

  • This study only assessed MRI measurements without clinical outcomes. So, there is no correlation analysis between the morphometric results and clinical results.

This is correct. The purpose of the study was only to describe the morphometric variations of the 3 tendons in order to assist ultimately in the preoperative autograft selection process.

  • For the improved understanding, figures measuring the parameters in MRI should be added for readers. 

Figure 1a and b was added (Line 115).

  • What is the clinical importance of this study? 

Using the described methodology, it becomes easier to avoid unnecessary tendon harvesting. For example, if the quadrupled hamstring tendon diameter will be less than 7 mm in diameter of if the patellar tendon is short or less than 20 mm in width then the preferred tendon source should be the quadriceps tendon.

  • Line 96-105. Please suggest references for the reasons why authors assessed in this way. 

Initially we started measuring the patellar tendon starting at the lower patellar pole and advancing 1 cm distally. In the process, we found out that due to variation in the patellar tendon length in men and women in most subjects there were 5 measurement sites while in shorter subjects only 4 measurement sites were available. In order to make the results comparable we decided to standardize the measurement. Thus, the proximal and distal attachment of the patellar tendon and two locations between them were selected.

  • Line 235. we performed evaluated..? 

Corrected. Thank you.

  • What are the limitations of this study?

The limitations are presented in lines 276-283.

Round 2

Reviewer 1 Report

Dear Authors, the manuscript has improved, I still have some small concerns:

In the abstract, the number of subjects included in your study is missing from the results

79 Design and Participants

80-81 the selection of 100 participants is a result… in the methods put the inclusion criteria and exclusion criteria…

144 Start the results section with a description of the participants included as line 80-81

158 Mean not Average

For the figures, I continue to recommend entering the standard deviation as a line above the bars. This is strictly necessary to understand the real evaluation of the measures

Author Response

Dear Reviewer,

Thank you for your valuable comments.

In the abstract, the number of subjects included in your study is missing from the results

The number of the participants (n=100) is written in line 13 and the number according to their sex is included. Due to limitation of the abstract length the results are not very detailed.

79 Design and Participants

Corrected as title of the Paragraph 2.1.

80-81 the selection of 100 participants is a result… in the methods put the inclusion criteria and exclusion criteria…

Improved, lines 79-80

144 Start the results section with a description of the participants included as line 80-81

Improved, lines 147-148

158 Mean not Average

Corrected.

For the figures, I continue to recommend entering the standard deviation as a line above the bars. This is strictly necessary to understand the real evaluation of the measures

Done. I missed to do it in the last correction. It is really important.

Reviewer 2 Report

Dear authors,

After revision process, this manuscript has been improved. 

However, this study describes only the morphometric MRI of three autografts without clinical results. This is the major limitation of this study.

As there are morphometric differences among theses grafts, the methods for graft preparation and fixation are different for each graft. 

However, this study simply compared morphometric characteristics of the tendon itself. 

To have clinical meaningfulness, post-operative MRI evaluations and clinical outcome measures are necessary.

Author Response

Dear Reviewer,

Thank you for your valuable comments. I have to note though that the purpose of the study was not to evaluate the clinical outcome of the postoperative MRI outcome of the various graft types but to show that the most abundant graft source around the knee joint is the quadriceps tendon and that the dimension of the tendon vary along their length. This autograft source can reliably provide adequately sized autograft material for ACL reconstruction. While considerable success in the restoration of knee stability has been demonstrated in ACL reconstruction, several recent studies indicate that between 1.8 and 22% of primary grafts will still fail globally and require revision if the hamstring autograft is less than 8mm in diameter. (Kamien PM et al. Am J Sports Med. 2013 Aug;41(8):1808-12, Murgier J et al.  Knee Surg Sports Traumatol Arthrosc. 2021 Mar;29(3):725-731,

Additionally, there are various MRI and ultrasound studied correlating the preoperative imaging evaluation of the hamstring tendons with the intraoperative measurements (Conte EJ et al. Arthroscopy. 2014 Jul;30(7):882-90). This has not been done for the quadriceps and the patellar tendons and is a paper we are currently working on.

Round 3

Reviewer 2 Report

Dear authors,

This study analyzed morphometric differences in 3 tendons used in ACLR.  Although this manuscript has been improved after revision procedures, this study has some fundamental limitations.

First, this study did not analyze clinical outcomes after ACLR using 3 different autografts.

Second, according to the type of grafts, preparation and fixation methods are different.

Third, after ACLR, tendon grafts undergo ligamentization process in the knee joint. During this ligamentization process, the quality of the grafts could be changed remarkably with time.

 Fourth, as authors indicated in the introduction section, donor site morbidity is an important issue in graft harvest.

However, there is lack of data and discussion about above mentioned issues. To have more clinical interests and importance, further experiments and analysis regarding such issues are needed.